# The prevalence and genomic context of Shiga toxin 2a genes in *E. coli* found in cattle

**Tomas Jinnerot[1], Angeles Tatiana Ponton Tomaselli[2], Gro Skøien Johannessen[2], Robert Söderlund[1], Anne Margrete Urdahl[2], Anna Aspán[1], Camilla Sekse[2]***

**1** National Veterinary Institute, Uppsala, Sweden, **2** Norwegian Veterinary Institute, Oslo, Norway

* Camilla.sekse@vetinst.no

## Abstract

Shiga toxin-producing *Escherichia coli* (STEC) that cause severe disease predominantly carry the toxin gene variant $stx_{2a}$. However, the role of Shiga toxin in the ruminant reservoirs of this zoonotic pathogen is poorly understood and strains that cause severe disease in humans (HUSEC) likely constitute a small and atypical subset of the overall STEC flora. The aim of this study was to investigate the presence of $stx_{2a}$ in samples from cattle and to isolate and characterize $stx_{2a}$-positive *E. coli*. In nationwide surveys in Sweden and Norway samples were collected from individual cattle or from cattle herds, respectively. Samples were tested for Shiga toxin genes by real-time PCR and amplicon sequencing and $stx_{2a}$-positive isolates were whole genome sequenced. Among faecal samples from Sweden, $stx_1$ was detected in 37%, $stx_2$ in 53% and $stx_{2a}$ in 5% and in skin (ear) samples in 64%, 79% and 2% respectively. In Norway, 79% of the herds were positive for $stx_1$, 93% for $stx_2$ and 17% for $stx_{2a}$. Based on amplicon sequencing the most common $stx_2$ types in samples from Swedish cattle were $stx_{2a}$ and $stx_{2d}$. Multilocus sequence typing (MLST) of 39 $stx_{2a}$-positive isolates collected from both countries revealed substantial diversity with 19 different sequence types. Only a few classical LEE-positive strains similar to HUSEC were found among the $stx_{2a}$-positive isolates, notably a single O121:H19 and an O26:H11. Lineages known to include LEE-negative HUSEC were also recovered including, such as O113:H21 (sequence type ST-223), O130:H11 (ST-297), and O101:H33 (ST-330). We conclude that *E. coli* encoding $stx_{2a}$ in cattle are ranging from strains similar to HUSEC to unknown STEC variants. Comparison of isolates from human HUS cases to related STEC from the ruminant reservoirs can help identify combinations of virulence attributes necessary to cause HUS, as well as provide a better understanding of the routes of infection for rare and emerging pathogenic STEC.

## Introduction

Shiga toxin-producing *Escherichia coli* (STEC) are zoonotic pathogens, occurring as abundant commensals among ruminants while occasionally causing gastrointestinal disease in humans. STEC infections can lead to the rare, but severe, hemolytic uraemic syndrome (HUS), with

**Data Availability Statement:** All raw data files are available from the ENA database (accession number(s) PRJEB35296 and PRJEB38743.

**Funding:** The work performed in Sweden was funded by the Swedish Board of Agriculture and the Swedish Civil Contingencies Agency. The work performed in Norway was funded by The Research Council of Norway (Project no 221663; Pathogens in the food chain) and the Norwegian Veterinary Institute. The Norwegian samples were collected in a survey funded by the Norwegian Food Safety Authority. The funders had no role in study design, data collection and analysis, decision to publish, or preparation of the manuscript.

**Competing interests:** The authors have declared that no competing interests exist.

children and the elderly being most at risk [1]. HUS can be fatal or lead to long-term sequelae with reduced kidney function or less commonly gastrointestinal or cognitive disabilities [2]. Though STEC constitute a genetically diverse group and all STEC should be considered potential agents of severe human disease [3], certain lineages appear to pose a far higher risk of causing HUS compared to other STEC. These STEC are referred to as HUSEC, i.e. *E. coli* previously associated with HUS [4]. In addition to the characteristics of the infecting STEC strain, the risk of developing HUS for a given patient is likely to be affected by host factors such as age and immunological status and possibly other factors like inoculum size and route of infection. Nonetheless, the identification of HUSEC strains is valuable for prioritizing interventions to reduce the exposure of humans to the most dangerous forms of STEC and to predict the progression of cases of illness.

STEC carry genes encoding Shiga toxins (Stx), considered to be their primary virulence factor. Stx genes are encoded by lambdoid bacteriophages that are maintained in a lysogenic stage in the bacterial chromosome [5]. Two types of Shiga toxins are known, Stx1 and Stx2, both of which are grouped into several subtypes. The presence of genes encoding a toxin variant referred to as Shiga toxin 2a (encoded by $stx_{2a}$ genes) has been repeatedly shown to be a trait shared by the majority of HUSEC strains [4,6,7]. However, while the presence of $stx_{2a}$ appears to be useful as a marker for the potential to cause HUS for strains infecting humans, the role of Shiga toxin in the ruminant reservoirs of STEC is poorly understood. Therefore, strains that cause severe disease in humans likely constitute a small and atypical subset of the overall STEC flora in ruminants. The prevalence and characteristics of major human pathogenic serotypes like O157:H7 that include known HUSEC have been extensively investigated in many countries over the last decades. Many other virulence-associated genes have been described and intimin (*eae*) is probably the principal adherence factor in human pathogenic STEC. The intimin gene is encoded on a pathogenicity island called the locus of enterocyte effacement (LEE), a genomic region encoding a system for attachment to the intestinal mucosa and translocation of effector proteins into host cells [3,8]. LEE is common in pathogenic STEC, but not essential for causing severe disease [9–11]. LEE-negative (*eae*-negative) STEC strains have also been associated with severe disease such as HUS, and they probably possess alternative mechanisms for attachment, such as *aggR* described for STEC/EAEC O104:H4 [12] and *saa* as described for STEC O113:H21 [13]. Few if any studies have been performed with the aim of providing an unbiased and comprehensive view of STEC strains carrying $stx_{2a}$ that occur in ruminant reservoirs. The aim of the present study was 1) to investigate the presence of $stx_{2a}$ genes in cattle samples collected in Norway and Sweden, 2) investigate the presence of $stx_2$ variants in cattle using high throuput amplicon sequencing and 3) characterize recovered $stx_{2a}$-positive *E. coli* and relate their phylogeny and virulence characteristics to known HUSEC.

## Materials and methods

### Development of a real-time PCR for $stx_{2a}$

A hydrolysis probe based real-time PCR assay for the detection of $stx_{2a}$ (Table 1) with primers containing locked nucleic acids (Exiqon A/S, Vedbaek, Denmark) was developed by alignment of 94 sequences of $stx_2$ representing all the recognised subtypes, 2a through 2g [14]. The novel assay was tested against a panel of *E. coli* strains obtained from the EU reference laboratory for *E. coli* (Istituto Superiore di Sanità, Rome, Italy) and from Statens Serum Institut (Copenhagen, Denmark) with known $stx_1$ and $stx_2$ subtypes. The assay was also tested in parallel with conventional PCR for $stx_2$ subtyping according to the reference laboratory [15] on a panel of STEC strains to verify the performance of the method (Table 2). PCR efficiency was assessed

**Table 1. Primers and hydrolysis probes used in this study for screening of Shiga toxin genes.**

| Designation | Sequence 5'-3' with modifications* | Usage | Reference |
|---|---|---|---|
| stx-F | TTTGTYACTGTSACAGCWGAAGCYTTACG | $stx_1$ and $stx_2$ real-time PCR | [16] |
| stx-R | CCCCAGTTCARWGTRAGRTCMACRTC | | |
| stx1-probe | CY5-CTGGATGATCTCAGTGGGCGTTCTTATGTAA-BHQ2 | | |
| stx2-probe | FAM-TCGTCAGGCACTGTCTGAAACTGCTCC-BHQ1 | | |
| VT2a-QfLNA | GGCGG+TTTT+ATT+TGCATTA+G | $stx_{2a}$ real-time PCR | This study |
| VT2a-QrLNA | CG+TC+AAC+CTT+CACTGT+A | | |
| VT2a-Qp | ATTO550-CRCAATCCGCCGCCATTGCATTAACAGAA-BHQ2 | | |
| F4_ad | *TCGTCGGCAGCGTCAGATGTGTATAAGAGACAG***GGCACTGTCTGAAACTGCTCCTGT** | $stx_2$ amplicon sequencing[a] | Modified from Persson et al. [6] |
| F4f_ad | *TCGTCGGCAGCGTCAGATGTGTATAAGAGACAG***CGCTGTCTGAGGCATCTCCGCT** | | |
| R1_ad | *GTCTCGTGGGCTCGGAGATGTGTATAAGAGACAG***ATTAAACTGCACTTCAGCAAATCC** | | |
| R1ef_ad | *GTCTCGTGGGCTCGGAGATGTGTATAAGAGACAG***TAAACTTCACCTGGGCAAAGCC** | | |

*Y = C or T, S = G or C, W = A or T, R = A or G, M = A or C

LNA, locked nucleic acid, as indicated by a preceding plus sign in the primer sequence.

[a] $stx_2$ specific sequences in bold and adapter sequences in italics.

by analysing serial dilutions of DNA from the positive control EDL933 in the Swedish lab. DNA concentration was measured using Qubit QuantIT HS kit (Invitrogen, Carlsbad, CA).

## Samples

As part of a nationwide prevalence survey of STEC in Swedish cattle, individual faecal and skin (ear) samples were collected at different slaughterhouses during 2011 to 2012. The total number of faecal and skin samples were 2041 and 418, respectively. Skin samples are likely to reflect bacteria previously shed by the individual animal as well as bacteria shed by other animals in the same group transferred via the environment or direct interaction e.g. grooming. They allow the efficient recovery of strains representative of a group of animals as they tend to be positive more often compared to individual faecal samples; the collection of ears for this purpose is easily standardized [17]. The number of samples collected at different slaughterhouses were determined to represent the geographical distribution of cattle in Sweden. In Norway, pooled faecal samples were collected through a nationwide survey sampling dairy herds with more than 50 cows in 2014 [18]. From each herd, faecal material was collected from ten different places, and were to include all present age groups. In total, samples were retrieved from 179 dairy herds.

## Sample preparation and isolation of $stx_{2a}$-positive *E. coli*

All samples were enriched in buffered peptone water at 37°C for 18–20 hours. After enrichment, DNA was extracted directly from the enrichment broths using QIAamp DNA Stool Mini kit (Qiagen, Hilden, Germany) according to the manufacturer's instructions. The extracted DNA was used for testing for the presence of Shiga toxin genes; $stx_1$, $stx_2$ and $stx_{2a}$ as described below. Isolation of $stx_{2a}$-positive *E. coli* was attempted from all $stx_{2a}$-positive samples, comprising plating onto different agar plates (i.e. MacConkey agar, Sorbitol MacConkey agar and/or CHROMagar™ O157) and incubation at 37°C overnight. Colonies with presumptive *E. coli* morphology were selected for further testing for presence of $stx_{2a}$ and DNA was extracted from colony material by boiling and tested by the $stx_{2a}$ real-time PCR. Presumptive $stx_{2a}$-positive *E. coli* were confirmed as *E. coli* using MALDI-TOF-MS (Bruker, Bremen,

**Table 2. Shiga toxin-producing *Escherichia coli* strains and other *E. coli* variants for exclusion and inclusion criteria using an $stx_{2a}$-specific real-time PCR.**

| | | | | Real-time PCR | Conventional PCR [15] | |
|---|---|---|---|---|---|---|
| | Type | ID | Origin/ Reference | $stx_{2a}$ | $stx_{2a}$ | Other *stx* types |
| Strains tested in the Swedish lab. | O26 | 35989 | cattle | + | + | *nd* |
| | O26 | 60409 | cattle | + | + | *nd* |
| | O26 | 56296 | human | + | + | *nd* |
| | O26 | 56299 | human | + | + | *nd* |
| | O26 | 11929 | human | + | + | *nd* |
| | O121 | 93404 | cattle | + | + | *nd* |
| | O121 | 93671 | cattle | + | + | *nd* |
| | O121 | 56304 | human | + | + | *nd* |
| | O121 | 61726 | human | + | + | *nd* |
| | O121 | 61734 | human | + | + | *nd* |
| | O157 | 7351 | cattle | + | + | *nd* |
| | O157 | 7353 | cattle | + | + | *nd* |
| | O157 | 7881 | cattle | + | + | $stx_{2c}$ |
| | SF-O157 | 73985 | human | + | + | *nd* |
| | O103 | 61704 | human | + | + | *nd* |
| | O103 | 6 | human | + | + | *nd* |
| | O104:H4 | 34474 | human | + | + | *nd* |
| | O157:H7 | EDL933 | [14] | + | + | $stx_{1a}$ |
| | O121 | 77173 | QA | + | + | *nd* |
| | O157 | 77180 | QA | + | + | *nd* |
| | O157 | 77185 | QA | + | + | *nd* |
| | O48:H21 | 81459 | QA | + | + | $stx_{1a}$ |
| | O103 | 23866 | QA | + | + | *nd* |
| | O26 | 77179 | QA | + | + | *nd* |
| | O26 | 14684 | cattle | - | - | *nd* |
| | O26 | 2733 | cattle | - | - | *nd* |
| | O157 | 492 | cattle | - | - | $stx_{2c}$ |
| | O157 | 55439 | cattle | - | - | $stx_{2c}$ |
| | O157 | 52063 | cattle | - | - | $stx_{2c}$ |
| | O103 | 61713 | human | - | - | *nd* |
| | O157 | CCUG 42744 | REF | - | - | $stx_{1a}$ and $stx_{2c}$ |
| | *E. coli* | ATCC 35218 | REF | - | - | *nd* |
| | O174:H21 | 81460 | QA | - | - | $stx_{2b}$ and $stx_{2c}$ |
| | O118:H12 | 81462 | QA | - | - | $stx_{2b}$ |
| | O73:H18 | 81463 | QA | - | - | $stx_{2d}$ |
| | unknown | 81464 | QA | - | - | $stx_{2g}$ |
| | ONT:H19 | 81465 | QA | - | - | $stx_{1d}$ |
| | unknown | 81466 | QA | - | - | $stx_{2f}$ |
| | O174:H8 | 81467 | QA | - | - | $stx_{1c}$ and $stx_{2b}$ |
| | O139:H1 | 81468 | QA | - | - | $stx_{2e}$ |
| Strains tested in the Norwegian lab | O113 | 51033 | minced meat | + | + | $stx_{1a}$ and $stx_{2dact}$ |
| | O157:H7 | EDL933 | [14] | + | + | $stx_{1a}$ |
| | O48:H21 | 94C | [14] | + | + | $stx_{1c}$ |
| | O146:H21 | NVI_257 | QA | + | + | - |

(*Continued*)

**Table 2.** (Continued)

| | Type | ID | Origin/ Reference | Real-time PCR | Conventional PCR [15] | |
|---|---|---|---|---|---|---|
| | | | | $stx_{2a}$ | $stx_{2a}$ | Other |
| | | | | | | $stx$ types |
| | O157:H7 | NVI_856 | QA | + | + | $stx_{2c}$ |
| | O26:H11 | 102–09817 | Human | + | + | - |
| | O103:H25 | 11060424 | Human | + | + | - |
| | O26:H11 | 1107–2514 | Human | + | + | - |
| | O26:H11 | 1107–2561 | Human | + | + | - |
| | O145:H28 | 11101865 | Human | + | + | - |
| | O145:H28 | 11111058 | Human | + | + | - |
| | O157:H7 | NVI-5768 | QA | + | + | $stx_{2c}$ |
| | O121:H19 | NVI-6694 | QA | + | + | - |
| | O157:H7 | NVI-8375 | QA | + | + | - |
| | O26:H11 | NVI-9669 | QA | + | + | - |
| | O145:H25 | 2013-22-83-1-2 | Sheep | + | + | - |
| | O121 | NVI-9529 | QA | + | + | - |
| | O145 | 000816 | Human | + | + | - |
| | O26:H11 | 2007-60-10067 | Sheep | + | + | - |
| | O26:H11 | 2007-60-10714 | Sheep | + | + | - |
| | O26:H11 | 2007-60-11809 | Sheep | + | + | - |
| | O174:H8 | DG131/3 | [14] | - | - | $stx_{1c}$ and $stx_{2b}$ |
| | O118:H12 | EH250 | [14] | - | - | $stx_{2b}$ |
| | O174:H21 | 31 | [14] | - | - | $stx_{2b}$ and $stx_{2c}$ |
| | O73:H18 | C165-02 | [14] | - | - | $stx_{2d}$ |
| | O139:H1 | S1191 | [14] | - | - | $stx_{2e}$ |
| | O128ac:H2 | T4/97 | [14] | - | - | $stx_{2f}$ |
| | O2:H25 | 7v | [14] | - | - | $stx_{2g}$ |
| | O171:H2 | NVI-136 | QA | - | - | $stx_{2b}$ and $stx_{2c}$ and $stx_{2d}$ |
| | O145:H34 | NVI-298 | QA | - | - | $stx_{2f}$ |
| | O139:H1 | NVI-427 | QA | - | - | $stx_{2e}$ |
| | O2:H25 | NVI-937 | QA | - | - | $stx_{2g}$ |
| | O22:H8 | NVI-949 | QA | - | - | $stx_{1c}$ and $stx_{2b}$ |
| | O91:H21 | NVI-967 | QA | - | - | $stx_{2d}$ |
| | O128:H- | E120 | Sheep | - | - | $stx_{1d}$ and $stx_{2b}$ |
| | O146:H21 | E382 | Human | - | - | $stx_{1d}$ and $stx_{2b}$ |
| | O166:H15 | NVI-2134 | QA | - | - | $stx_{2d}$ |
| | O113:H4 | NVI-3748 | QA | - | - | $stx_{1c}$ and $stx_{2b}$ |
| | O146:H21 | NVI-9954 | QA | - | - | $stx_{2d}$ |
| | O157 | 2014-01-5652 | Cattle | - | - | $stx_{2c}$ |

QA, quality assurance test strains obtained from the European reference laboratory for *E. coli* (Istituto Superiore di Sanità, Rome, Italy) or Statens Serum Institut (Copenhagen, Denmark). REF, reference strain. nd, not done.

Germany) and characterized by WGS as described below. Extracted DNA from a subset of cattle skin (ear) samples were subjected to high-throughput amplicon sequencing of $stx_2$ genes.

## Real-time PCR detection of *stx* genes

DNA extracts from all the enriched samples were subjected to screening by real-time PCR for the detection of $stx_1$ and $stx_2$ as described by Perelle *et al.* 2004 [16] and ISO/TS 13136:2012 [19] as well as for $stx_{2a}$ by the novel real-time PCR assay (Table 1). The PCR analyses were performed slightly differently in Sweden versus in Norway; In the Swedish lab, PCR was performed in 15 μL PCR-reaction volumes containing 333 nM of each primer, 100 nM of probe, 2 μL template DNA and PerfeCTa qPCR Toughmix with Low ROX (Quantabio, Beverly, MA). PCR was conducted with an ABI 7500 Fast thermocycler (Life Technologies, Carlsbad, CA) using the following thermal profile: 95˚C for 3 minutes followed by 45 cycles of 95˚C for 3 seconds and 60˚C for 30 seconds at which fluorescence was measured. Samples were regarded as positive if they resulted in a Ct-value <45. In the Norwegian lab, PCR was performed in 20 μL PCR-reaction volumes containing 333 nM of each primer, 100 nM of probe, 5 μL template DNA and 10 μL Brilliant III Ultra-Fast QPCR Master Mix (Agilent Technologies, Santa Clara, CA). PCR was conducted with a Stratagene Mx3005P (Agilent technologies) using the following thermal profile: 95˚C for 3 minutes followed by 45 cycles of 95˚C for 3 seconds and 60˚C for 30 seconds at which fluorescence was measured. Samples were regarded as positive if they resulted in a Ct-value <45.

## High-throughput amplicon sequencing of *stx₂* genes from skin samples

To further estimate the prevalence of $stx_2$ variants and combinations of variants, the sequencing protocol developed by Persson and co-workers [6,14] was adapted to the deep amplicon sequencing protocol provided by Illumina [20]. The modified primers are shown in Table 1. This assay was applied to DNA extracted from enrichment broths of a subset of 48 $stx_2$ PCR-positive Swedish cattle skin (ear) samples. Amplicons generated using the modified primers were barcoded using Nextera XT index kit (Illumina, San Diego, CA). An Illumina MiSeq instrument was used for sequencing 250 base pairs in both directions. Primer sequences were removed and reads were quality trimmed using Trimmomatic 0.32 [21]. Any read pair with less than 200 bp of high quality sequence from each direction was discarded. Read pairs were mapped against 89 known $stx_2$ variants [14], and the closest match for each read pair was determined. Samples were discarded if less than 100 read pairs were consistent with $stx_2$, and any sequence variant supported by <5% of the total reads for a given sample was ignored to remove spurious variants generated by sequencing errors. With the minimum accepted length of $2 \times 200$ bp, all known variants could be unambiguously classified as either $stx_2$ subtype a, b, c, d, e, f or g with the exception of the sequence variants AF500189 ($stx_{2d}$, could not be distinguished from certain $stx_{2c}$) and M59432 ($stx_{2c}$, could not be distinguished from certain $stx_{2a}$) which occurred in zero and one sample respectively.

## Statistical analysis

Shiga toxin gene prevalence with 95% confidence intervals were calculated by the exact binomial test using R version 3.6.2 for Windows [22].

## Whole genome sequencing of *stx₂ₐ*-positive *E. coli* isolates

DNA was extracted from $stx_{2a}$-positive *E. coli* isolates with QIAmp DNA mini kit or using an EZ1 Biorobot with the DNA Tissue kit (Qiagen, Hilden, Germany). The Nextera XT Kit

(Illumina, San Diego, CA) was used to prepare libraries. The Swedish isolates were sequenced on a MiSeq (Illumina, San Diego, CA) as paired-end 2×250 base pair reads and the Norwegian isolates were sequenced on a HiSeq 2500 instrument using rapid mode generating paired-end reads from each isolate in two lanes i.e. 2×2×250 base pair reads. All isolates were sequenced to >30 × depth. Quality control was performed using FastQC and sequence data was uploaded to the European Nucleotide Archive under project numbers PRJEB35296 and PRJEB38743 for the Swedish and Norwegian data respectively. Reads were trimmed and draft assemblies were generated using SPAdes 3.5.0.

### Genomic characterization of $stx_{2a}$-positive *E. coli* isolates

The phylogroup of the $stx_{2a}$-positive *E. coli* isolates was determined by *in silico* implementation of the PCR system of Clermont et al. [23] on assembled genomes. Multilocus sequence typing (MLST) [24], *in silico* serotyping [25] and virulence gene detection [26] was performed using Centre for Genomic Epidemiology web services [27]. Clustering of MLST data was performed using the minimum spanning tree algorithm in Bionumerics 7.6 (Applied Maths NV, Sint-Martens-Latem, Belgium), with each locus as an equivalent categorical variable. Multiple correspondence analysis was performed on virulence gene presence/absence data in R 3.5.0 [22] using the FactoMineR library.

## Results

### Development of a real-time PCR for $stx_{2a}$

The specificity of the novel PCR assay was confirmed at both laboratories by analyzing clinical and reference strains encoding $stx_1$ and/or $stx_2$. All 45 $stx_{2a}$-positive strains and none of the 35 $stx_{2a}$-negative strains were identified as $stx_{2a}$-positive by the real-time PCR (Table 2). The PCR efficiency was calculated to be 75% based on the standard curve from serial dilutions of DNA from EDL933.

### Detection of *stx* gene variants in cattle samples

Table 3 presents the prevalence, with their corresponding 95% confidence intervals, of the various *stx* genes in samples from the two cattle surveys in Sweden and Norway, respectively. In short, for skin (ear) samples taken in the Swedish survey the prevalence was 64.1%, 78.7% and 1.6% for $stx_1$, $stx_2$ and $stx_{2a}$, respectively. The corresponding results for the faecal samples in Sweden were 37.2% for $stx_1$, 52.9% for $stx_2$ and 5.0% for $stx_{2a}$. The herd prevalence detected from the pooled faecal samples in Norway were 79.3%, 93.9% and 16.8% for $stx_1$, $stx_2$ and $stx_{2a}$, respectively.

Amplicon sequencing of a subset of Swedish $stx_2$ PCR-positive skin broth samples produced high-quality output for 31 of 48 samples, revealing a high number of samples being

**Table 3. Prevalence of Shiga toxin genes (with 95% CI) in Swedish dairy cattle and Norwegian dairy herds using different sampling procedures.**

| Sample | Total | $stx_1$ | | $stx_2$ | | $stx_{2a}$ | |
|---|---|---|---|---|---|---|---|
| | | *n* | % (95% CI) | *n* | % (95% CI) | *n* | % (95% CI) |
| Sweden | | | | | | | |
| skin | 418 | 268 | 64.1 (59.3–68.7) | 329 | 78.7 (74.5–82.5) | 7 | 1.6 (0.7–3.4) |
| faecal | 2041 | 760 | 37.2 (35.1–39.4) | 1081 | 52.9 (50.8–55.1) | 103 | 5.0 (4.1–6.1) |
| Norway | | | | | | | |
| pooled faecal | 179 | 142 | 79.3 (73.0–85.0) | 168 | 93.9 (89.3–96.9) | 30 | 16.8 (11.6–23.1) |

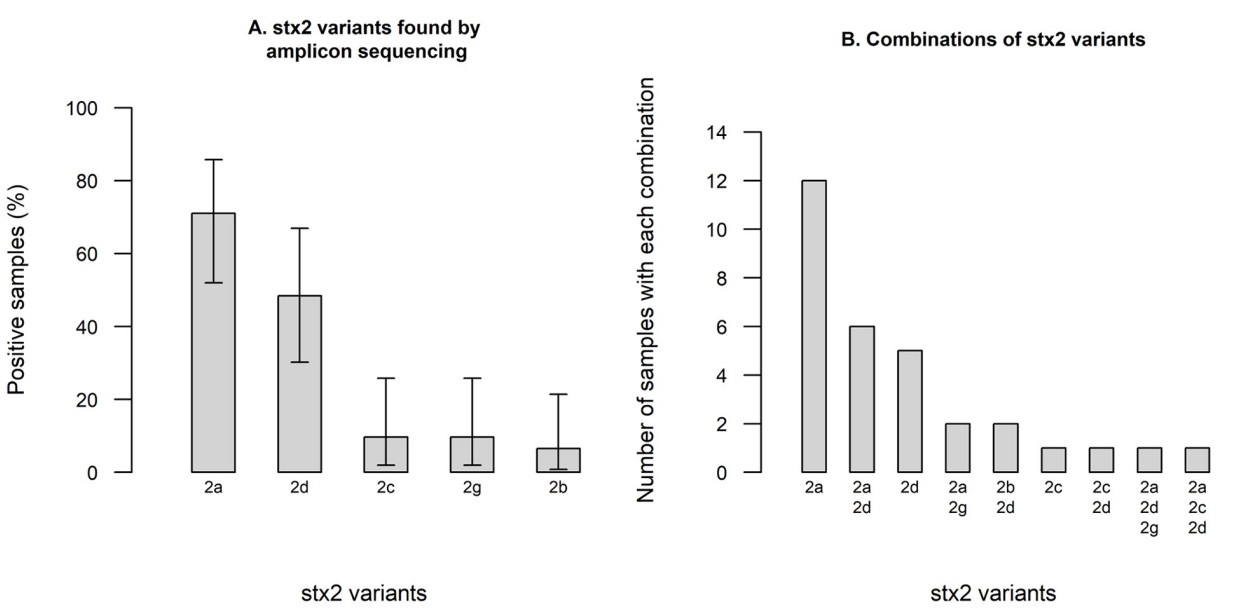

**Fig 1. Prevalence of Shiga toxin 2 variants in 31 $stx_2$-positive skin samples from Swedish cattle evaluated by amplicon sequencing.** The figure shows (A) the prevalence, with 95% confidence intervals, of each $stx_2$ variant and (B) combinations of different toxin variants found in individual samples.

positive for $stx_{2a}$ (71.0%) and $stx_{2d}$ (48.4%), with a lower proportion of samples positive for $stx_{2c}$ (9.7%), $stx_{2g}$ (9.7%), and $stx_{2b}$ (6.5%) (Fig 1A). Most samples were positive for only a single $stx_2$ variant, but several were positive for two or three different variants, most commonly $stx_{2a}$ together with $stx_{2d}$ (Fig 1B). Only four out of the 22 samples in which $stx_{2a}$ was identified by amplicon sequencing were positive for $stx_{2a}$ when tested by real-time PCR.

## Genomic characterization of $stx_{2a}$-positive *E. coli* isolates

In total, 40 $stx_{2a}$-positive *E. coli* isolates were retrieved from the Swedish (n = 25) and the Norwegian (n = 15) samples by plating the enriched samples onto selective agar plates and selecting one isolates per positive sample. One isolate was determined not to be $stx_{2a}$-positive after genome analysis and excluded from further analysis, bringing the total number of included isolates to 39. Characteristics of the $stx_{2a}$-positive *E. coli* isolates are summarized in Fig 2 and S1 Table. Most of the isolates belonged to phylogroup B1 (n = 27), but A (n = 8) and B2 (n = 3) were also represented. One isolate could not be assigned to a phylogroup as it presented an undefined profile (+/-/+/+). A total of 19 MLST profiles were identified, of which 11 profiles formed three clonal complexes (CC) centered around ST-10 (phylogroup A, CC10, 9 isolates), ST-223 (phylogroup B1, CC155, 7 isolates) and ST-718 (phylogroup B1, 10 isolates from different predefined and undefined CC's). These three CC's were recovered from both Swedish and Norwegian cattle (Fig 3). Isolates were assigned to a CC if they differed from one another by a maximum of three loci. *In silico* serotyping revealed substantial diversity but was generally in agreement with MLST and phylogroup divisions (Fig 2). Virulence gene detection revealed five isolates to be "classical" LEE-positive STEC with intimin (*eae*), *tir*, *tccP* and a repertoire of non-LEE encoded effector genes. These isolates were O121:H19 (ST-655), O26:H11 (ST-21),

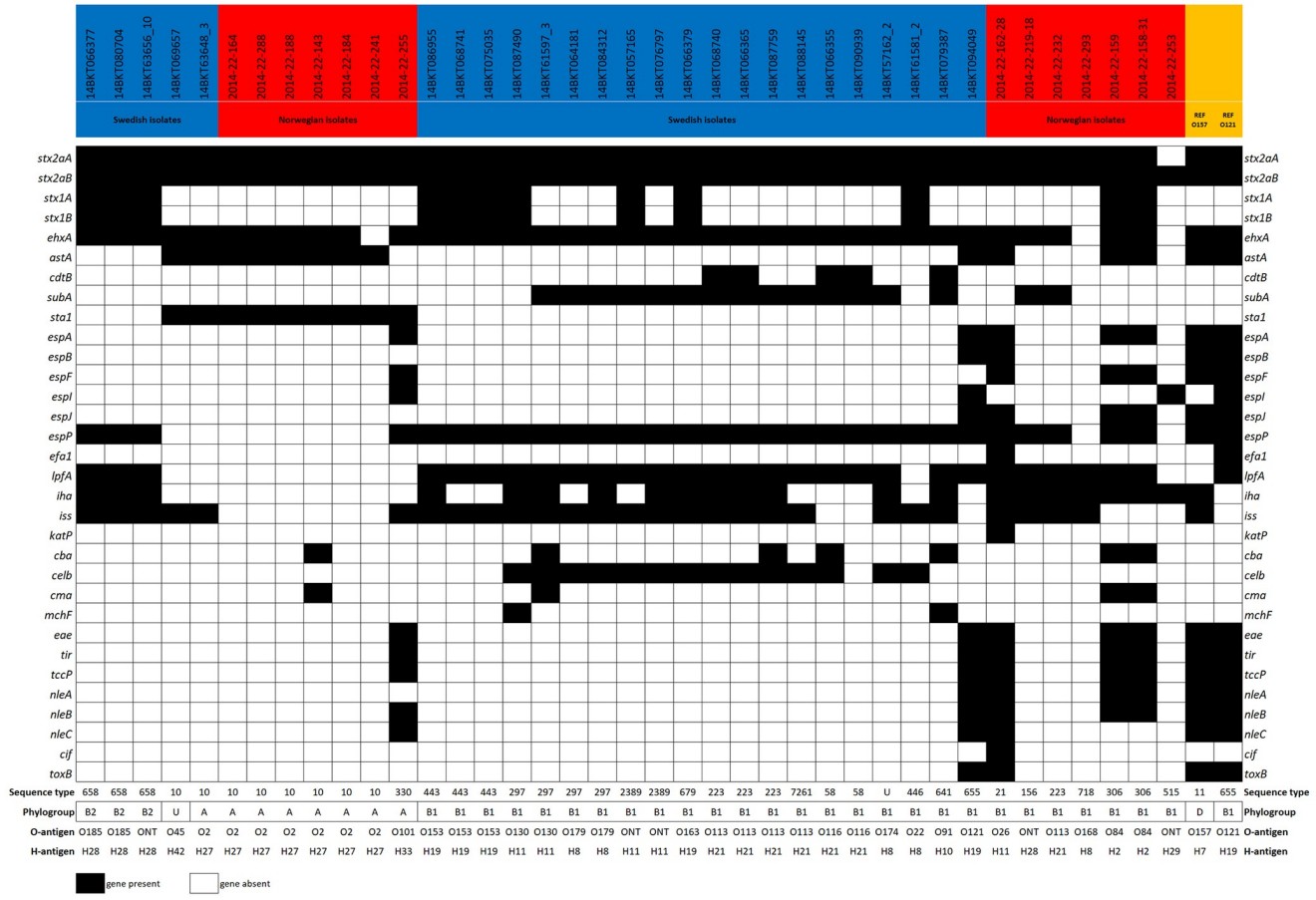

**Fig 2. Virulence gene profiles, sequence types, phylogroups and serotypes of 39 *stx*<sub>2a</sub>-positive *E. coli* isolates from cattle in Sweden and Norway.** Two outbreak strains of EHEC (O157:H7 TW14359 and O121:H19 16–9255) are included as references. U; undetermined phylogroup or sequence type. ONT; antigen non-typable. O2 not distinguishable from O50. O101 not distinguishable from O162. O153 not distinguishable from O178.

O84:H2 (ST-306, two isolates) and O101:H33 (ST-330). Notably, 36 of the 39 isolates were positive for haemolysin genes (*ehxA*). Multiple correspondence analysis supported the presence of a distinctive group of LEE-positive isolates and a second group of STEC/ETEC (enterotoxigenic *E. coli*) hybrids with heat-stable enterotoxin 1 (*sta*₁) but lacking *lpfA* and *espP* among other traits (Fig 4). The latter group corresponded to the previously mentioned CC10 clonal complex. A single isolate in this complex (O101:H33 ST-330) was intermediate in being both LEE-positive and an STEC/ETEC hybrid.

## Discussion

### Prevalence of *stx*₂ and *stx*₂ₐ in cattle

Shiga toxin-producing *E. coli* are generally considered a normal part of the healthy ruminant intestinal flora, so the high prevalence of both *stx*₁ and *stx*₂ genes observed among cattle in the present study is not surprising. As some STEC seem to be more associated with severe human infection than others, the general detection of *stx* genes without any knowledge of the subtypes present is not well suited for detecting severe human pathogens. The *stx* genes are encoded in prophage genomes, and free phage particles might increase the load of PCR-detectable *stx*

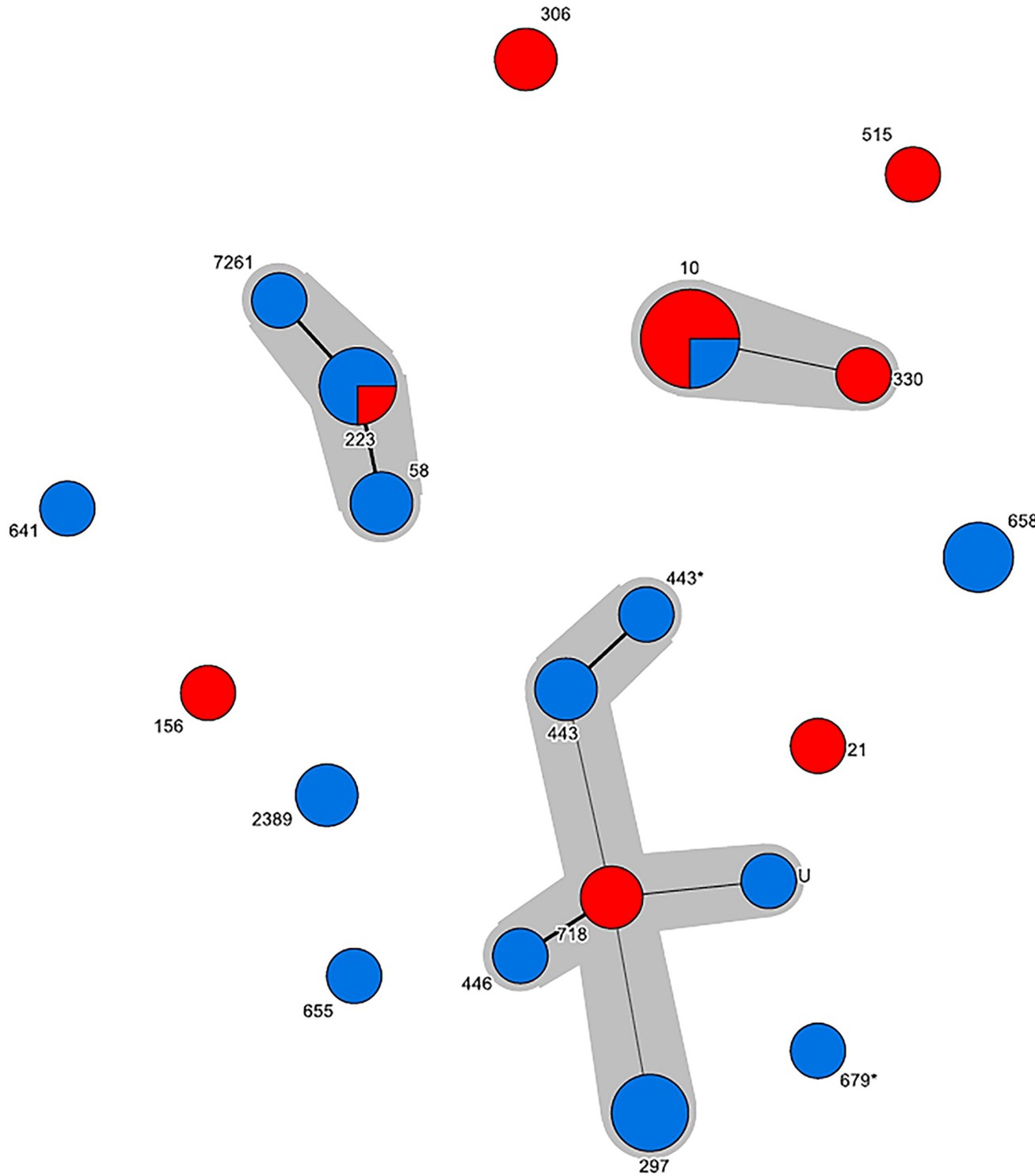

**Fig 3. Minimum spanning tree of sequence types among $stx_{2a}$-positive *E. coli* from cattle.** All three major clonal complexes comprise isolates of both Swedish (blue) and Norwegian (red) origin. A difference in only one SNP is displayed by a thick line whereas a difference in two SNP's is displayed by a thin line.

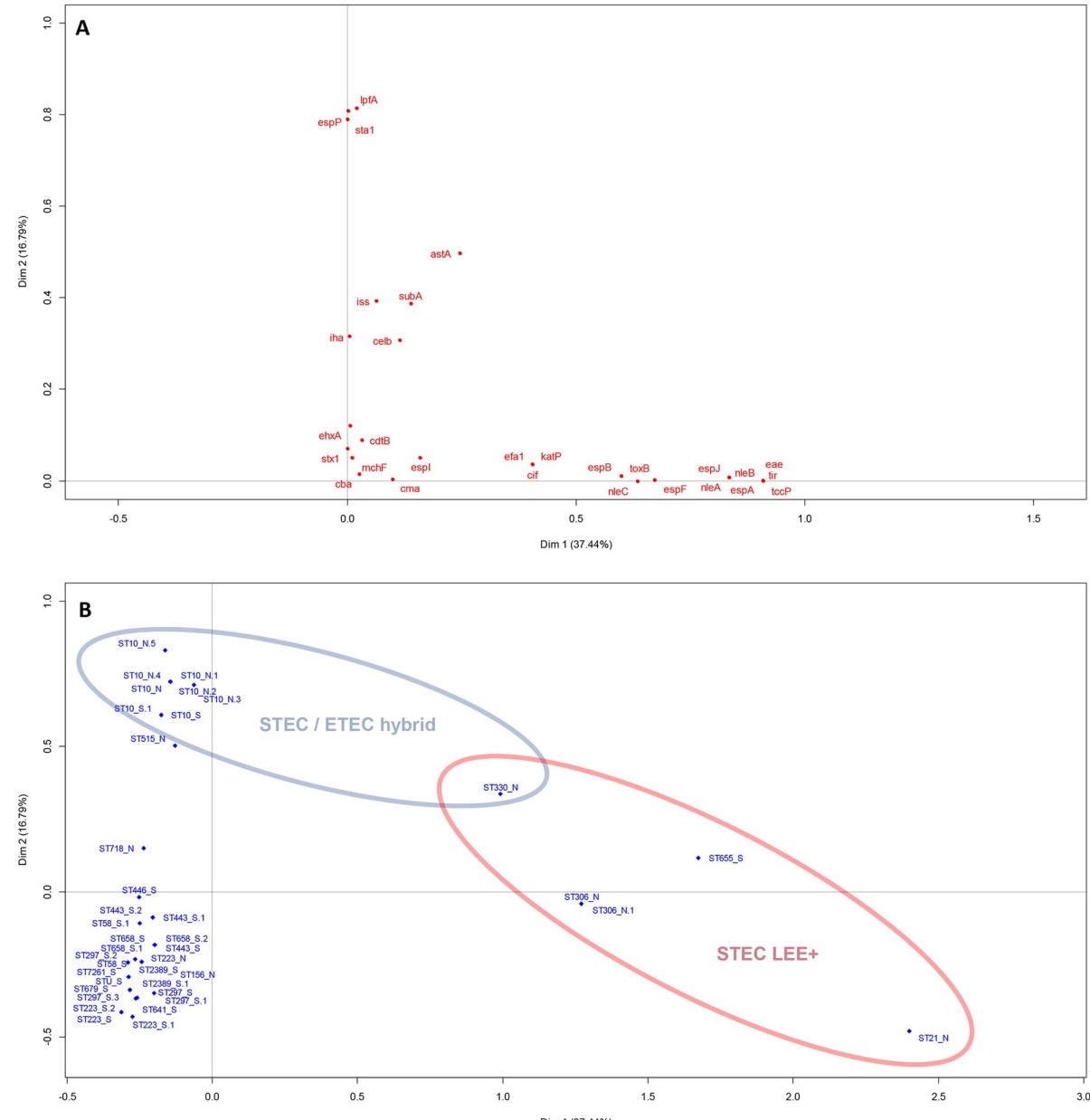

**Fig 4. Multiple correspondence analysis biplots of the virulence genes of *stx*₂ₐ-positive *E. coli* from Norwegian and Swedish cattle.** Association between virulence genes is estimated based on co-occurence in isolates (A), and association between isolates is estimated based on the genes they possess or lack (B).

genes in environments where *stx*-carrying bacteria are present. Both shedding and colonization status for ruminants with STEC can be expected to be periodic or transient; the presence of potential pathogens on the herd level can therefore be more relevant compared to the prevalence in individual animals.

In the Swedish survey, the skin (ear) samples resulted in higher prevalence's than the corresponding faecal samples. This is in concordance with previous results from Sweden [17], and indicate that skin (ear) samples represents the prevalence in an animal herd or group and not individual animals. The Norwegian survey analyzed pooled faecal samples collected from dairy herds, giving herd prevalence and not individual which is probably the reason these resulted in the highest $stx$ prevalence. Due to the differences in sampling and study design, the $stx$ prevalence in the two countries cannot be compared directly. This was not the purpose of this study.

In recent years, studies have shown that STEC strains carrying $stx_2$ are more pathogenic than those carrying only $stx_1$, while STEC strains with the $stx_{2a}$ subtype are most frequently associated with severe STEC disease manifestation; HUS [4,6,7]. The association between $stx_{2a}$ and HUS made this subtype a particular focus of the presented study. As more STEC strains have been investigated by whole genome sequencing over the years it has been shown that there are sequence diversity within the $stx_{2a}$ subtype, with some variants being highly similar to sequence variants of the $stx_{2c}$ subtype [14]. In the present study, a novel real-time PCR assay was developed to specifically detect $stx_{2a}$ genes. This PCR was employed for primary screening of samples in order to identify samples for attempting isolation of $stx_{2a}$-positive $E.\ coli$ isolates from the two countries for further characterization. As specificity was the primary concern in the development of this PCR, the sensitivity is likely to be suboptimal and the PCR level prevalence of $stx_{2a}$ reported for the different sample types should be interpreted with caution. The poor sensitivity was a necessary trade-off to produce a sufficiently specific assay for identifying $stx_{2a}$-positive samples due to the high sequence similarity of $stx_{2a}$ compared to other $stx_2$ subtypes. The real-time PCR described here has successfully been employed in two different laboratories using different PCR reagents and platforms which indicates that the method is robust and can easily be used in different laboratories. As the real-time PCR has not been tested on spiked samples with known levels of $stx_{2a}$-positive STEC we do not know how the Ct values relates to number of cells. We are, however, aware that a Ct-value of 45 might be a false positive. In the case of $stx_{2a}$-positive STEC, we consider it more important to avoid false negative samples than having a few false positive samples.

A community profiling approach based on amplicon sequencing of partial $stx_2$ genes was used on a subset of the $stx_2$ PCR-positive samples. This revealed $stx_{2a}$ and $stx_{2d}$ to be the most common types of $stx_2$ among Swedish cattle samples, but also the presence of several other variants and combinations of variants. With these results in mind, identifying all potential STEC in a sample, especially an environmental or primary production sample, one would need to retrieve and test several isolates. This is in line with recommendations in ISO/TS-13136:2012 [19]. It is also reasonable to believe that one or more $stx_1$-subtypes, in addition to several $stx_2$-subtypes, could be present in a sample as well as different strains with the same $stx_1$- or $stx_2$-subtype, e.g. $stx_{2a}$ could be present in two different genomic backbones in the same sample. However, this was beyond the scope of this study. The approach of using amplicon sequencing described in this study is a powerful tool for detecting all known $stx_2$ variants in a sample, only limited by the inclusivity of the standard $stx_2$ sequencing primers. A comparison between the real-time PCR and amplicon sequencing results revealed that only four out of 22 samples positive for $stx_{2a}$ by amplicon sequencing were also positive for the same subtype by PCR. Although the limit of detection for the PCR was low when evaluated using purified DNA from a defined strain this might not be the case for mixed samples where other subtypes of $stx_2$ may interfere with the amplification. However, there might also be an actual difference in the analytical sensitivity between the two methods. The real-time PCR was primarily used in this study as a screening tool in order to retrieve $stx_{2a}$-positive $E.\ coli$ and therefore the discrepant results were not further investigated.

## Characteristics and public health relevance of LEE-positive STEC isolates

Most STEC associated with HUS reported in the literature to date have carried $stx_{2a}$ genes in combination with the locus of enterocyte attachment and effacement (LEE) [28]. In the present study, only one LEE-positive isolate was detected from the Swedish samples; an O121:H19. STEC O121:H19 is a well-known cause of severe outbreaks and sporadic infections, and is a known HUSEC [4,29]. From the Norwegian samples one LEE-positive STEC of serotype O26:H11 was detected. This serotype is known to carry different variants of $stx$ genes or to be $stx$-negative when isolated from ruminants, and $stx_{2a}$-positive strains of O26:H11 have caused several cases of HUS in Norway [30] as well as elsewhere [31]. Two isolates of O84:H2 (ST-306) with LEE and harbouring both $stx_1$ and $stx_{2a}$ genes were also recovered from the Norwegian samples. O84:H2 has been found in cattle and as a cause of sporadic cases of diarrhoea among humans in New Zealand [32], but has to our knowledge not been linked to severe cases of illness. However, strains of the same sequence type, ST-306, with different serotypes have been linked to HUS cases in both Germany [4] and Sweden [33].

## Characteristics and public health relevance of LEE-negative STEC isolates

LEE-negative STEC are rare, but perhaps underestimated as a cause of HUS, as diagnostics have historically focused on the most well-known LEE-positive serotypes. In general, LEE-negative isolates rely on alternative host tissue adhesion mechanisms, which are either poorly understood or known, but historically associated with other *E. coli* pathotypes [9–11]. The most notable example of LEE-negative HUSEC is the major outbreak in Europe 2011 of enteroaggregative *E. coli* O104:H4 encoding $stx_{2a}$ [34], but smaller outbreaks and sporadic cases are continuously reported. Two variants of LEE-negative STEC recovered from Swedish cattle in the present study, O113:H21 (ST-223) and O130:H11 (ST-297), have caused sporadic cases of HUS in Sweden [35]. O113:H21 (ST-223) is a well-known LEE-negative HUSEC with cases reported from several countries [36], while O130:H11 has also been associated with HUS in Australia [37] and Argentina [9]. The presence of both of these strains in Swedish and Norwegian cattle thus should be considered a public health concern. Another potential HUSEC was an O163:H19, a serotype that has previously been linked to a case of HUS in the UK [38]. The remaining serotypes and STs found in the present study have not to our knowledge, been linked to severe illness in humans. However, several known LEE-negative HUSEC could not be distinguished from strains not known to cause HUS in terms of virulence gene repertoire (Fig 4), this might be due to the selection of genes included in this study which is biased towards LEE-positive STEC and other known pathotypes. Previous studies have had similar difficulties in identifying the virulence determinants of non-LEE HUSEC [39], but new potentially relevant markers are continuously being discovered [40].

## STEC/ETEC hybrid isolates

In recent years, hybrid pathotypes of *E. coli* have been reported to be associated with diarrhea and HUS [32,39–42]. Several isolates obtained in this study carried genes encoding heat-stable enterotoxin 1 ($sta_1$) in addition to $stx_{2a}$ and can thus be considered hybrids between the STEC and ETEC (enterotoxigenic *E. coli*) pathotypes. The isolates appear to be related to some extent, belonging to phylogroup A and ST-10, but belonging to multiple serotypes with O2/O50:H27 being the most common. This lineage was found in isolates from both Norway and Sweden, and STEC/ETEC hybrids of matching serotypes have previously been isolated from patients and cattle in Italy and Finland [41,43]. A single isolate of ST-330 in the present study had both a LEE-region, the ETEC $sta_1$ toxin and belonged to the same clonal complex as the ST-10 STEC/ETEC LEE-negative hybrids. A strain matching the description of the present

finding (O101:H33, phylogroup A, ST-330, $sta_1$+) have been associated with HUS in an infant in Finland in 2001 [44]; another isolate with matching phylogroup and ST, but no expressed serotype (ONT:H⁻), has been reported from a German HUS case [4].

## Conclusions

In this study, we found a substantial proportion of $stx_{2a}$-positive samples from Swedish and Norwegian cattle. $stx_{2a}$ and $stx_{2d}$ were the most common variants among Swedish cattle samples analyzed by high throuput amplicon sequencing, however other variants and combinations of variants were also seen. This approach of using amplicon sequencing is a powerful tool for detecting all known $stx_2$ variants in a sample and reflects the importance of selecting more than one $stx$-positive isolate in a complex sample. Isolation and characterization of 39 $stx_{2a}$-positive *E. coli* revealed that only a small proportion have similar virulence profiles as known HUSEC-strains and only a few known HUSEC lineages were identified among the $stx_{2a}$-positive isolates. We conclude that $stx_{2a}$-positive *E. coli* in cattle are ranging from strains similar to HUSEC to unknown STEC variants. It is currently unclear whether most $stx_{2a}$-positive *E. coli* pose a risk of causing HUS in a vulnerable patient, or if this capability is an emergent property of combinations of other virulence factors including key adhesins. Due to this unexplored field it would be of interest to compare human and animal isolates further by comparative WGS analysis.

## Supporting information

**S1 Table. Virulence genes in $stx_{2a}$ isolates.**
(XLSX)

## Acknowledgments

The sequencing service for the Norwegian isolates was provided by the Norwegian Sequencing Centre (www.sequencing.uio.no), a national technology platform hosted by the University of Oslo and supported by the "Functional Genomics" and "Infrastructure" programs of the Research Council of Norway and the Southeastern Regional Health Authorities.

## Author Contributions

**Conceptualization:** Tomas Jinnerot, Gro Skøien Johannessen, Robert Söderlund, Anne Margrete Urdahl, Anna Aspán, Camilla Sekse.

**Data curation:** Tomas Jinnerot, Robert Söderlund, Camilla Sekse.

**Formal analysis:** Tomas Jinnerot, Angeles Tatiana Ponton Tomaselli, Robert Söderlund.

**Funding acquisition:** Anne Margrete Urdahl, Anna Aspán.

**Investigation:** Tomas Jinnerot, Angeles Tatiana Ponton Tomaselli, Robert Söderlund.

**Methodology:** Tomas Jinnerot, Camilla Sekse.

**Supervision:** Gro Skøien Johannessen, Camilla Sekse.

**Visualization:** Robert Söderlund.

**Writing – original draft:** Tomas Jinnerot, Camilla Sekse.

**Writing – review & editing:** Angeles Tatiana Ponton Tomaselli, Gro Skøien Johannessen, Robert Söderlund, Anne Margrete Urdahl, Anna Aspán.

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
