## [Decision Letter · Decision Letter 0]

22 May 2020

PONE-D-20-09947

The prevalence and genomic context of Shiga toxin 2a genes in *E. coli * found in cattle

PLOS ONE

Dear Dr. Sekse,

Thank you for submitting your manuscript to PLOS ONE. After careful consideration, we feel that it has merit but does not fully meet PLOS ONE’s publication criteria as it currently stands. Therefore, we invite you to submit a revised version of the manuscript that addresses the points raised during the review process.

The manuscript has been well received by the reviewers, though there are some remarks that need to be addressed. Please go over all of them and adapt where appropriate.

We look forward to receiving your revised manuscript.

Kind regards,

Patrick Butaye, DVM, PhD

Academic Editor

PLOS ONE

Journal Requirements:

Reviewers' comments:

Reviewer's Responses to Questions

**Comments to the Author**

1. Is the manuscript technically sound, and do the data support the conclusions?

Reviewer #1: Yes

Reviewer #2: Yes

2. Has the statistical analysis been performed appropriately and rigorously? 

Reviewer #1: Yes

Reviewer #2: No

3. Have the authors made all data underlying the findings in their manuscript fully available?

Reviewer #1: Yes

Reviewer #2: Yes

4. Is the manuscript presented in an intelligible fashion and written in standard English?

Reviewer #1: Yes

Reviewer #2: Yes

5. Review Comments to the Author

Reviewer #1: Well-written and interesting paper, only a few minor concerns

L24 skin samples needs more explanation - as initially guessed rectal biopsy

L28 and elsewhere in manuscript - Need to be careful describing isolates as HUSEC. Your cattle isolates might be similar to those which have caused HUSEC, but at present still no absolute way of determining which isolates will cause human disease.

Would instead use "strains similar to HUSEC" to make this distinction. If we could perfectly identify HUSEC, would not be a need for your study, but as stated on L 33 we are working towards being able to identify HUSEC (not there yet).

L30 ST=sequence type? Needs to be defined

L31 same problem as on L28

L45 previously associated with HUS

Table 1 Some interesting nucleotides in these primers. Only familiar with A, C, T, and G. What are Y, W, and R? More description needed here.

L101 Having so much stx appearing in the ear is interesting. A bit more discussion about the use of the ear for sampling would be welcome. We have taken swabs of various parts of bovine anatomy to find STEC, but never considered the ear.

L326 Would suggest deleting 'finding that also represent a'

Reviewer #2: PONE-D-20-09947

The authors describe a study carried out in Sweden and Norway to assess the prevalence of shiga-toxin genes of clinical importance in cattle. While the study is of interest I do not think that in its current form it is not suitable for inclusion in PLOS one without considerable revision.

Introduction – Good introduction using relevant reference.

Methods – Authors have used appropriate methods for analysing the sample tested.

Line 120- further details of skin sampling required.

Line 137- was any attempt made to confirm that the sensitivity of these assays.

Line 139- why adopt different methods for analysing different samples- explain.

Results

You have used a ct value of <45 to indicate a positive sample. Do you know how this relates to cell numbers in this sample, which will contain a mixture of DNA from many different bacterial species present in the sample?

Is there agreement between the levels of stx2a detected by amplicon sequencing and RT-PCR?

Line 168 missing reference -xxxx

Line 199-201 makes me concerned about the sensitivity of this RT-PCR do you have another explanation.

Line 213 It is likely that the low recovery rate relates to the method used as if performed as described in the methods it is likely that many different E. coli and other bacteria will be growing on the plates and identifying the stx positive ones very difficult.

Figure 1a error bars are higher than the axis

Figure 2 can’t read poor quality.

Discussion

Line 269 remove either

Line 279 Please explain why sensitivity is not important?

Line 289 it’s essential to also consider that although stx gene are detected in a sample they may not be associated with E. coli. Consider the presence of free-phage.

Line 320 has any attempt been made to compare the genomes of these isolates if available.

Line 330 consider using other databases to identify virulence determinants if the one used is bias towards LEE. Did you search you genomes for genes discussed in ref 41.

Line 350 how do you determine significance.

6. PLOS authors have the option to publish the peer review history of their article (what does this mean?). If published, this will include your full peer review and any attached files.

Reviewer #1: No

Reviewer #2: No

---

## [Author Response · Author response to Decision Letter 0]

2 Jul 2020

A copy of the reply to reviewers can be found as a uploaded file, but also pasted in here;

Reply to reviewers

Thank you for comments and suggestions for improving the manuscript. We have tried to answer each comments as good as we can in this response to the reviewers comments.

Reviewer #1: Well-written and interesting paper, only a few minor concerns

L24 skin samples needs more explanation - as initially guessed rectal biopsy

Answer: We have added (ear) to the description of skin samples in the abstract and explained about the skin samples in the materials and methods (L105-109).

L28 and elsewhere in manuscript - Need to be careful describing isolates as HUSEC. Your cattle isolates might be similar to those which have caused HUSEC, but at present still no absolute way of determining which isolates will cause human disease.

Would instead use "strains similar to HUSEC" to make this distinction. If we could perfectly identify HUSEC, would not be a need for your study, but as stated on L 33 we are working towards being able to identify HUSEC (not there yet).

Answer: We totally agree with the reviewer that naming animal isolates as human pathogenic or HUSEC as there is not possible to determine whether they are true human pathogenic or not. We have changed from LEE-positive HUSEC to “LEE-positive strains similar to HUSEC”.

L30 ST=sequence type? Needs to be defined

Answer: We have defined ST as sequence type in L30.

L31 same problem as on L28

Answer: We have changed to “Lineages known to include LEE-negative HUSEC were also recovered including, such as O113:H21 (sequence type (ST)-223), O130:H11 (ST-297), and O101:H33 (ST-330).” in L29-L31 and to ”strains similar to HUSEC” in L32. 

L45 previously associated with HUS

Answer: We have added “previously” to L46.

Table 1 Some interesting nucleotides in these primers. Only familiar with A, C, T, and G. What are Y, W, and R? More description needed here.

Answer: These nucleotides are degenerated bases and represent two binding alternatives and not only one as for A, C, T, G. We have included a description of the degenerate base symbols as a note to Table 1, L92. The codes used are standard as defined by IUPAC. 

L101 Having so much stx appearing in the ear is interesting. A bit more discussion about the use of the ear for sampling would be welcome. We have taken swabs of various parts of bovine anatomy to find STEC, but never considered the ear.

Answer: Faecal shedding of STEC seems to be intermittent and often limited to a fraction of the animals in a group. In contrast, skin samples will likely represent an “archive” of recent shedding from the individual animal and other animals in the group. In O157:H7 studies we have found far more positive samples using ears compared to faeces collected in parallel. Therefore, we consider skin samples a cost-effective way of sampling if the goal is to collect as many isolates as possible rather than determining the individual prevalence. Various rubbing, grooming, eating, drinking etc. behaviours are likely to lead to bacteria being transferred to the head of the animal. Compared to other types of skin samples, ears are easy to collect in a standardized manner for the regular slaughterhouse staff. We have expanded the relevant part of the materials and methods section explaining this (L105-L109).

L326 Would suggest deleting 'finding that also represent a'

Answer: We agree and have deleted this and changed from suspected to “potential HUSEC”

Reviewer #2: PONE-D-20-09947

The authors describe a study carried out in Sweden and Norway to assess the prevalence of shiga-toxin genes of clinical importance in cattle. While the study is of interest I do not think that in its current form it is not suitable for inclusion in PLOS one without considerable revision.

Introduction – Good introduction using relevant reference.

Methods – Authors have used appropriate methods for analysing the sample tested.

Line 120- further details of skin sampling required.

Answer: We have added some details regarding the ear sampling and why this is done. (L105-L109) See also the response to Reviewer 1 above.

Line 137- was any attempt made to confirm that the sensitivity of these assays.

Answer: There was not performed spiking experiments with known number of Stx2-encoding bacteria to confirm the sensitivity of these assays. We are also aware that it might not be accurate enough to estimate limit of detection from the standard curve based on serial dilutions from a positive control, so we suggest to rewrite the following;

L87; “PCR efficiency was assessed by analysing serial dilutions of DNA from the positive control EDL933 in the Swedish lab”.

L189-L191; “The PCR efficiency was calculated to be 75% based on the standard curve from serial dilutions of DNA from EDL933.”

As we already stated in the discussion, this real-time PCR was primarily used as a screening tool to retrieve stx2a-positive E. coli.

Line 139- why adopt different methods for analysing different samples- explain.

Answer: The method is mainly the same, but we are using different qPCR mastermix. The reason for that is because the two different labs usually prefer the mastermix used and therefore have stocks of this available. We also recognize that we have used different sample volumes which might have an impact on the detection. As this might be of importance for assessing the PCR results. Although we have only tested the method in two laboratories using different PCR reagents and platforms, the results indicate that the method is robust and can easily be used in different laboratories. We have included a paragraph regarding this in the discussion (L290-293). 

Results

You have used a ct value of <45 to indicate a positive sample. Do you know how this relates to cell numbers in this sample, which will contain a mixture of DNA from many different bacterial species present in the sample?

Answer: Since spiking experiments with known levels of stx2a has not been carried out, we do not know how the Ct-value relates to the number of cells in this PCR. We are, however, aware that a Ct-value at 45 might be a false positive. In the case of STEC we consider it more important to avoid false negatives than having a few false positive sample. We have included this in the discussion (L293-296).

Is there agreement between the levels of stx2a detected by amplicon sequencing and RT-PCR?

Answer: Unfortunately, neither method is well suited for quantitative analysis. The real-time PCR is designed to be specific at the cost of sensitivity and has low amplification efficiency. This is due to theoretical constraints from the high sequence similarity of stx2a to other stx2-subtypes. For amplicon sequencing we can exactly count the number of occurrences of each variant in the sample after PCR amplification and library generation, but this amplification is carried out using degenerate primers and is unlikely to have the same efficiency for different stx subtypes. We discussed using digital PCR, but again the low efficiency of the stx2a PCR would likely be a problem, in this case for endpoint signal thresholding. 

Line 168 missing reference –xxxx

Answer: The reference in L168 is now added; PRJEB38743.

Line 199-201 makes me concerned about the sensitivity of this RT-PCR do you have another explanation.

Answer: As mentioned above and in the manuscript itself, the sensitivity of the PCR is indeed likely to be suboptimal. This is not a design flaw but a necessary trade-off given the small differences between stx2-subtypes; it was originally designed to verify isolates as stx2a-positive. Here we also use it to screen samples for selecting those with a decent chance of isolating stx2a-positive strains by randomly selecting E. coli colonies. For this purpose specificity is again more important as we are only interested in strongly positive and true positive samples. It is therefore not surprising that the amplicon sequencing assay picks up more positive samples overall in this case, although we do not believe it is more sensitive as a detection technique in general. 

Line 213 It is likely that the low recovery rate relates to the method used as if performed as described in the methods it is likely that many different E. coli and other bacteria will be growing on the plates and identifying the stx positive ones very difficult.

Answer: The low recovery rate could definitively be due to high E. coli diversity and mainly other E. coli growing on the plates and therefore very difficult to identify the stx2a-positive bacteria. Another alternative might be colony hybridization with a specific probe, however, due to background noise this might not be a better option. Colony hybridization might require a copy of the plates for later selecting the stx positive colonies and would probably be more time consuming.

Figure 1a error bars are higher than the axis

Answer: The first author has access to the original figure 1a and are unfortunately on vacation at the moment. However, as soon as he is back we will upload a corrected Figure 1a. Hope that is acceptable for editor and the reviewer.

Figure 2 can’t read poor quality.

Answer: To the authors’ knowledge the quality of figure 2 is of sufficient quality when downloading the figure separately. The figure have been uploaded to PACE (https://pacev2.apexcovantage.com/) before submission to ensure that the figure meets PLOS One requirement. However, we have uploaded the original figure 2 which is in a wrong file format to see if that could help regarding the quality.

Discussion

Line 269 remove either

Answer: We have removed “either”.

Line 279 Please explain why sensitivity is not important?

Answer: We have added a sentence further explaining why sensitivity could not be prioritized (L288-290). 

Line 289 it’s essential to also consider that although stx gene are detected in a sample they may not be associated with E. coli. Consider the presence of free-phage.

Answer: This is mentioned in the first paragraph of the discussion as a possible reason for PCR-positive samples. However, free phages are likely more of a concern in studies where samples are directly analysed by PCR, in the present study all samples are pre-enriched amplifying prophage stx genes. 

Line 320 has any attempt been made to compare the genomes of these isolates if available.

Answer: Unfortunately we do not have access to genome data from the human isolates.

Line 330 consider using other databases to identify virulence determinants if the one used is bias towards LEE. Did you search you genomes for genes discussed in ref. 41?

Answer: To the authors’ knowledge virulence databases are difficult to interpret as many genes are virulence-associated or fitness genes, but not enough to cause disease alone. The people curating such databases need to have in depth knowledge about virulence genes and to our knowledge we are not aware of any virulence databases covering these new potential virulence-associated genes. We did not test specific virulence-associated genes as those referred to in ref. 41.

Line 350 how do you determine significance.

Answer: ”Significant” in a colloquial sense; we have now replaced the word with “substantial”.

---

## [Editor Report · Decision Letter 1]

21 Jul 2020

The prevalence and genomic context of Shiga toxin 2a genes in *E. coli * found in cattle

PONE-D-20-09947R1

Dear Dr. Sekse,

We’re pleased to inform you that your manuscript has been judged scientifically suitable for publication and will be formally accepted for publication once it meets all outstanding technical requirements.

Kind regards,

Patrick Butaye, DVM, PhD

Academic Editor

PLOS ONE